# Multi-proton dynamics near membrane-water interface

Subhasish Mallick ⓘ & Noam Agmon ⓘ ✉

Protons are crucial for biological energy transduction between membrane proteins. While experiments suggest rapid proton motion over large distances at the membrane-water interface, computational studies employing a single excess proton found the proton immobilized near the lipid headgroup. To address this discrepancy, we conduct DFTB3 simulations by incrementally adding protons up to three. We show that a single proton moves rapidly toward the nearest headgroup, where it is either repelled by a choline group or binds covalently to phosphatic oxygen. With multiple protons, while some are trapped by the lipid headgroups, the remaining proton diffuses laterally faster than in bulk water. Driven by excess energy, this proton initially jumps to the center of the water slab before relaxing into the third- and second-hydration shells. Lateral diffusion rates increase as the proton stabilizes in the second hydration shell. These results provide insights into proton dynamics near membranes and explain experimental observations.

Bioenergetics, i.e., cellular metabolism, involves energy transduction across disparate distance and timescales[1]. In both photochemical and oxidative phosphorylation, energy is first trapped by rapid transport along an electron transport chain, and subsequently converted to chemical energy by attaching an inorganic phosphate ($P_i$) to adenosine diphosphate, to produce adenosine triphosphate (ATP)[2]. Until 1960, it was thought that the electronic energy is converted to a hypothetic energy-rich species responsible for the ATP synthesis. But in 1961 Mitchell proposed[3] that there is an important interim step, in which the electronic energy is transformed into a proton gradient across a biological membrane (e.g., in the mitochondria). This became the basis of his "chemiosmotic theory". It involves two types of membrane-bound enzymes. One functions as a proton pump (e.g., cytochrome c oxidase, or complex IV in the eukaryotic mitochondria), while the other comprises of ATP-synthase, a rotary motor harnessing this protonic gradient to synthesizing ATP[4]. The Nobel Prize in Chemistry of 1978 was awarded to Peter D. Mitchell "for his contribution to the understanding of biological energy transfer through the formulation of the chemiosmotic theory".

According to Mitchell, the protonic electrochemical potential gradient is delocalized between the bulk aqueous phases on the two sides of the membrane. That is, each side of the membrane is under a state of "quasi-equilibrium". In contrast, Williams suggested that the

excess protons were localized on the membrane surface[1]. Experimentally, the pH does not decrease in bulk water near proton-pumping bacterial cells[5]. It was suggested that the negatively charged lipids were preventing protons from leaving the membrane surface[6].

In 1994, an experimental proof was supplied to the localized scenario by Heberle et al.[7], who studied proton transfer (PT) through the bacterial proton-pump bacteriorhodopsin (BRho), embedded in a piece of the purple membrane (the plasma membrane of *Halobacterium halobium*) of 600 nm diameter. Their results suggested that two-dimensional proton diffusion is faster than proton release to the bulk, supporting the localized, non-equilibrium scenario of Williams[1]. Similar experimental verifications soon followed[8–11].

In one experiment, cytochrome and ATP-synthase proteins were reconstituted in the membrane of either a small or a giant vesicle[12], in which the interprotein distance was estimated to be 80 nm and 10 $\mu m$, respectively. In Mitchell's chemiosmotic theory the protons are equilibrated with the bulk, so there should be no effect of the interprotein distance on the rate of ATP synthesis. However, the rate was found to be notably larger for the smaller distance, suggesting that the protons move exclusively along the membrane, at least up to 80 nm.

The most straightforward explanation for the confinement of proton fluxes to the membrane surface would be the existence of an energy barrier[13]. For example, one group[14] envisioned the barrier as

The Fritz Haber Research Center, Institute of Chemistry, The Hebrew University of Jerusalem, Jerusalem, Israel. ✉e-mail: agmon@fh.huji.ac.il

arising from a $H_3O^+$ ion sandwiched between two lipid headgroups, $P - O^- \cdots H_3O^+ \cdots {}^-O - P$ (Fig. 3 there). This suggests that the high barrier exists not only perpendicular to the membrane, but also laterally. Consequently, proton lateral diffusion would be slow, contradicting the BRho experiments[7]. Resolving this requires more quantitative experiments for determining the rates of both proton diffusion and its escape to the bulk.

Subsequent experiments, from Peter Pohl's lab, focussed on synthetic planar membranes without the proton-pump or ATP-synthase[15,16]. These membranes were enriched with molecules that release protons upon UV excitation (a localized "pH-jump" process), and with the pH sensor fluorescein monitoring proton arrival. Surprisingly, this gave diffusion constants nearly as large as for protons in bulk liquid water (ca. $9 \times 10^{-5}$ cm²/s), sometimes even larger[17]. Furthermore, these were independent of the membrane charge, suggesting proton motion in the membrane's hydration layer. Although Springer et al.[16] insisted that protons were not carried along by mobile buffers, or as $OH^-$, they have used a theoretical model[18] that assumes exactly that, resulting in poor agreement with experiment.

A simpler model[17] assumed two-dimensional proton diffusion at the interface, with a slow irreversible leak to the bulk (rate constant $k$). This agreed quantitatively with the experimental data (see Fig. 1 in ref. 17). It was also suggested that the simultaneous release of many protons saturates the protonable groups on the surface, while the remainder diffuse laterally unimpeded[17], their escape to solution restricted by an entropic barrier[19].

Unlike the Grotthuss mechanism in liquid water[20], which was confirmed on the atomic level by various flavours of quantum molecular dynamics (MD)[21], few simulations of protons at the lipid-water interface were conducted. One of the pioneering simulations applied multi-state empirical valence bond (MS-EVB2) for one excess proton near a DMPC membrane[22]. It found that "proton diffusion is significantly reduced as the proton penetrates into the polar region of the lipid membrane", forming Zundel cations which bridge together two lipid headgroups via strong hydrogen-bonds (no proton covalent bonding is allowed in this methodology). A subsequent simulation[23] further concluded that the hydronium is mostly attached to a lipid headgroup, thus diffusing with the characteristic lipid self-diffusion coefficient namely, much slower than in bulk liquid water.

The MS-EVB simulations[23] show that one or more of the hydronium's (Eigen or Zundel forms) O-H moieties are hydrogen-bonded to the lipid's phosphatic or carbonyl groups, facing away from the water phase. This is quite the opposite of a hydronium at the air-water interface, whose hydrogens point towards the water phase[24]. Moreover, in the latter case the proton attraction to the interface is weak, 1 $k_BT$ (experimentally, 1.3 kcal/mol)[25], whereas the binding to the membrane according to the MS-EVB simulation amounts to 5 kcal/mol. The common denominator between such protons at aqueous interfaces is an orientation that maximizes the hydrogen-bonding interactions.

Recently, ab initio MD (AIMD) simulations within the Car-Parrinello Molecular Dynamics (CPMD) framework were performed for one excess proton at the water-oil[26] and water-lipid interfaces[27]. Due to the heavy demands in computer time, these simulations[27] were confined to a small membrane patch (up to 10 lipids), and 10 ps duration. When the proton was initially placed in the interfacial water layer, it diffused among adjacent water molecules via the Grotthuss mechanism[20], eventually hopping across a water wire to an adjacent phosphate ion. However, although CPMD does allow protonation by covalent bond formation, this did not occur[27].

How can one explain the discrepancy between the experimentally fast lateral proton mobility vs. its immobilization at an interfacial phosphate in MD simulations? Yamashita and Voth[23] suggested that "some protons are trapped at negative sites of the membrane and prevent other protons from approaching these sites in the low pH experiment ... that might have measured the diffusion of such mobile

protons, but not the trapped protons". Similarly, Agmon and Gutman[17] proposed that a "proton front" is formed at the release site, which protonates the titratable sites, enabling the remaining protons to move unhindered on the surface. To our knowledge, such multi-proton simulations were not yet attempted.

In this work, we use Self-Consistent-Charge Density-Functional Tight-Binding with third-order energy correction (DFTB3)[28], for simulating a hydrated membrane patch occupied by one, two or three protons. Consistent with previous simulations, we observe that a single excess proton near the membrane rapidly migrates toward a nearby phosphate headgroup via a short water wire. However, for the first time, we report the formation of a PO-H covalent bond. In the presence of multiple protons, those near membrane leaflets bind and remain bound throughout the simulation, while additional protons diffuse laterally in the bulk at accelerated rates within the second or third hydration layer. Thus, we systematically expose how the trajectory of one proton profoundly influences the others, resolving the inconsistencies between experiment and theory.

## Results

### Initial conditions

In the final DFTB3 equilibrated geometry for a hydrated POPC membrane (see Methods), we replaced one, two, or three water molecules with hydronium ions ($H_3O^+$) at different positions. Then we performed DFTB3 production runs at constant temperature, volume and particle number (NVT) with a timestep of 0.5 fs, saving coordinates each timestep, for at least 10 ps each. The initial placements of the hydronium ions (Fig. 1) were as follows:

Simulation 1: A single excess proton in two different locations: (a) In the first hydration layer of the lipid, O $\cdots$ O distance approximately 2.8 Å from the nearest non-etheric phosphatic oxygen ($H_3O^+ \cdots O - PO_3^-$). (b) Approximately three hydration layers from the nearest phosphate group.

Simulation 2: Two excess protons in two different locations: (a) Oxygen-oxygen distance between the two $H_3O^+$ ions approximately 9.2 Å. The shortest $H_3O^+ \cdots O - PO_3^-$ distance was 2.1 Å, placing the ions in the first and third hydration layers of the phosphate group. (b) One proton covalently bound to a phosphatic oxygen, the second about 6.0 Å from it.

Simulation 3: Three excess protons in two different arrangements: (a) Starting from the initial configuration in Simulation 2a, a third $H_3O^+$ ion was added near the upper leaflet of the membrane. The initial O $\cdots$ O distance between this proton and the one in the bulk was ≈10.6 Å. (b) The water pool here was expanded by one additional layer, see Methods for details. Two $H_3O^+$ ions were positioned near the two opposing leaflets of the membrane, with the third in the bulk water region. This simulation is important for testing whether the mobile proton is diffusing in the center of the water pool or in the membrane hydration layers. The initial coordinates for all simulations are given in the Supplementary Data 1 file.

### Single Proton

Our first two DFTB3 simulations involve only one excess proton each, which we denote by $H_a^+$. We calculate the proton indicator[29] and track its distance as a function of time from various lipid atoms. In Simulation 1a, the hydronium was initially near a phosphatic oxygen atom, $O_P(i)$. Within a few femtoseconds (fs), $O_P(i)$ underwent protonation by $H_a^+$. Because $H_a^+$ was not yet equilibrated, it promptly (within 400 fs) escaped to a distance of 6 to 7 Å from the membrane surface (green line in Fig. 2a). It then engaged in lateral diffusion parallel to the membrane until about 2 ps. Then it abruptly approached the membrane, protonating another phosphate group, $O_P(1)$ (red line in Fig. 2a). This ultrafast transition was facilitated by a water-wire, $OW_1 \rightarrow OW_2 \rightarrow OW_3$ (Inset 1, Supplementary Movie 1.mp4), whose formation was also observed in an AIMD simulation[27].

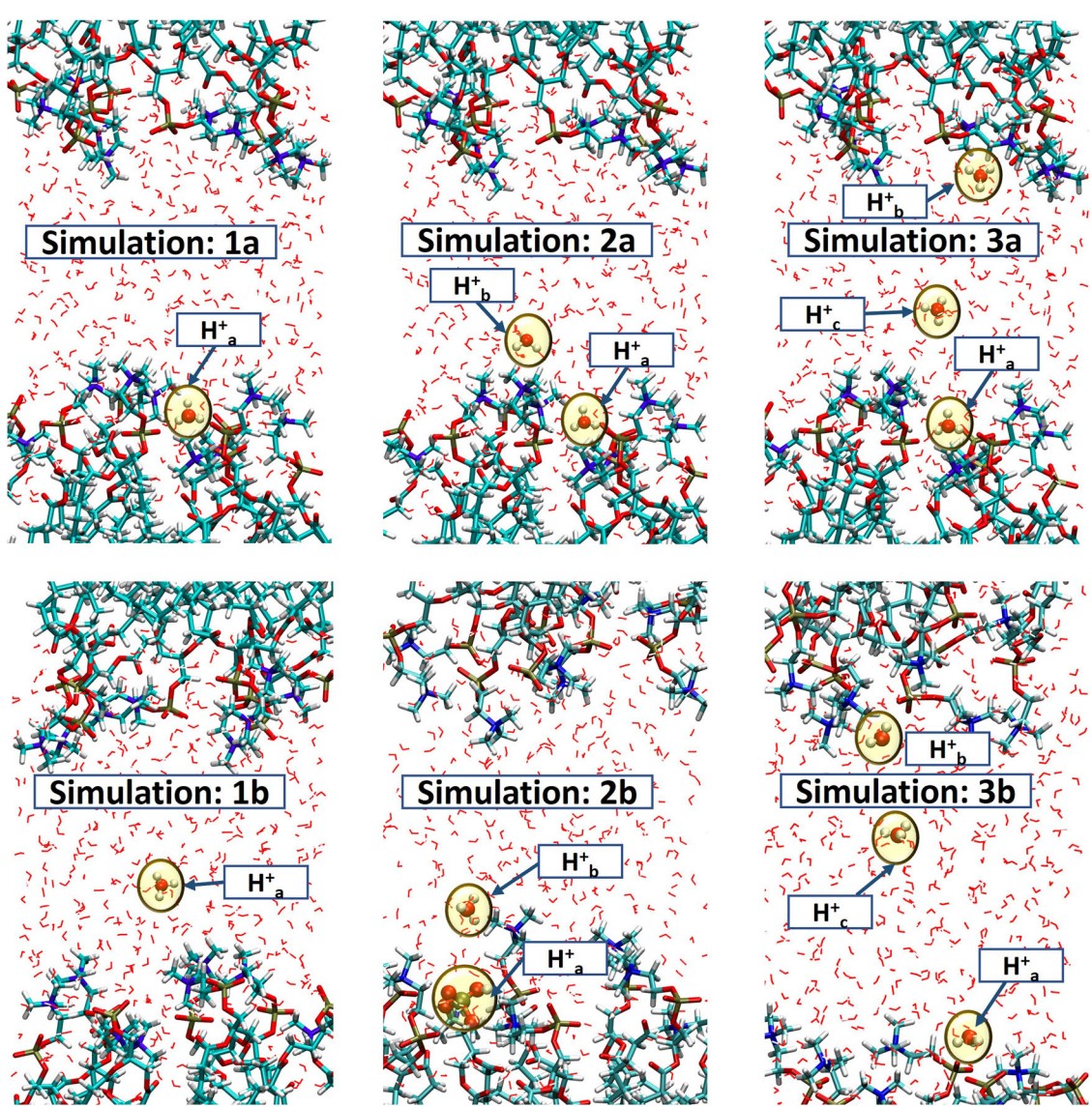

**Fig. 1 | Initial configurations for our six DFTB3 simulations.** (1a) Single proton placed near the membrane; (1b) Proton placed in the bulk. (2a) Two excess protons, one near the lower leaflet and the other in the bulk; (2b) One protonates a phosphate group and the second in the bulk; (3a) Three protons, two near the two opposing membrane leaflets, and one in the bulk. (3b) Similar to 3a, with an extra layer of water. Water molecules are represented as lines, lipids as licorice-style structures, and hydronium ions are highlighted yellow.

The $H_a^+ - O_P(1)$ bond, which persisted up to 8.2 ps, is a covalent bond, as evidenced by the short H–O distance of 1 Å and its minimal fluctuations. Interestingly, at ca. 6 ps an $O_P(2)$ formed a 2 Å long hydrogen-bond with the hydroxyl, $O_P(1) - H_a^+ \cdots O_P(2)$ (violet line). This set the stage for the direct PT from $O_P(1)$ to $O_P(2)$ at 8.2 ps, forming $O_P(1) \cdots H_a^+ - O_P(2)$. These two types of phosphate protonation events, from aqueous $H_3O^+$ or a protonated phosphate, were not reported by previous simulations.

To test whether the covalent bond formation is unique to DFTB3, we have carried out a Born-Oppenheimer molecular dynamics (BOMD) simulation (using the CP2K package) starting from the same two-proton initial configuration as Simulation 2a (Simulation 2a′ below). One of these protons binds covalently to a phosphate oxygen atom for the duration of the simulation, as observed in our DFTB3 simulations.

These phosphatic PT events can also be elucidated by monitoring the change in the partial Mulliken charge of the relevant phosphatic oxygens. Indeed, Fig. 2b shows that, during protonation of an $O_P$ atom, there is a noticeable increase in its partial charge. For example, around 2 ps protonation occurred at $O_P(1)$, causing an increase in its charge from approximately −0.45 to −0.3 a.u. Subsequently, around 8.2 ps, positive charge shifted from $O_P(1)$ to $O_P(2)$, commensurate with the PT event seen in Fig. 2a.

In Simulation 1b, the excess proton was initially about three hydration layers (ca. 10 Å) from the nearest $O_P$ of the lower leaflet (Fig. 3c, blue line). It is more convenient to measure this distance as the distance $Z$, perpendicular to the membrane, whose center is at $Z = 0$. The membrane width is then twice the average $Z$ coordinate of the P atoms, denoted $<Z_P>$. With $<Z_P> \approx 20$ Å, the initial proton location in Simulation 1b was thus $Z_{H^+} = 30$ Å (Fig. 3a, yellow line). It then moved, in ca. 40 fs, to $Z_{H^+} \approx 32.5$ Å, roughly midway between the two leaflets. This ultrafast motion was likely enabled by the proton's initial excess energy, propelling it during the first 2 ps further toward the upper leaflet. Then it suddenly reverted, returning to the bulk (see Supplementary Movie 2.mp4). This might be attributed to electrostatic repulsion from a nearby positively charged choline group, because at 2 ps the distance between $H_a^+$ and its nearest choline hydrogen ($H_C$) equalled to that from $O_P$ (4 Å), see Fig. 3c. The return to the bulk continued until 7 ps, and then the proton settled into lateral diffusion

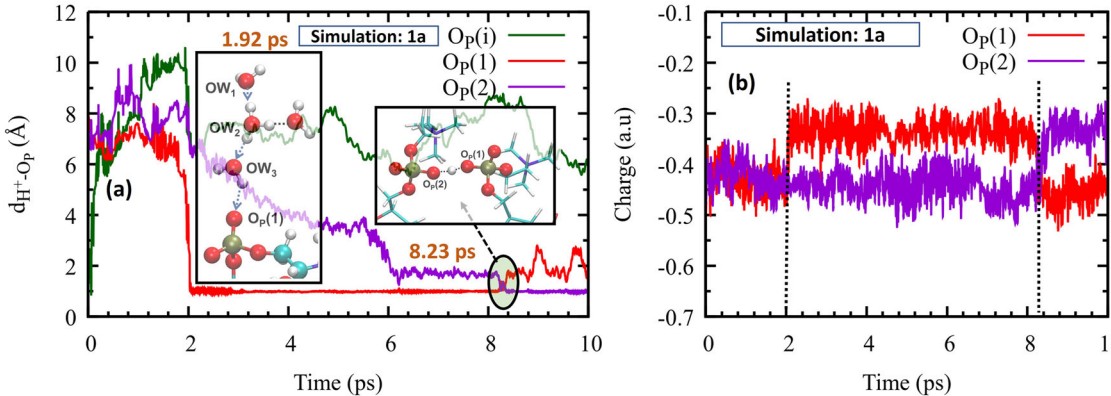

**Fig. 2 | Highlights from Simulation 1a. a** The time-dependent distances of the excess proton ($H_a^+$) from its initially nearest phosphatic oxygen ($O_P(i)$, in green), and from the two nearby phosphatic oxygens ($O_P(1)$, in red; $O_P(2)$, in violet) that participate in protonation events. Inset 1: A snapshot at 1.92 ps reveals an extended water-wire (blue dotted arrows) facilitating the protonation of $O_P(1)$. Inset 2:

A snapshot at 8.23 ps captures the transition state for PT from $O_P(1)$ to $O_P(2)$ (black circle). **b** The partial Mulliken charges of $O_P(1)$ (in red), and $O_P(2)$ (in violet) corroborate the two PT events occurring at the vertical dotted lines. Source data are provided as a Source Data file.

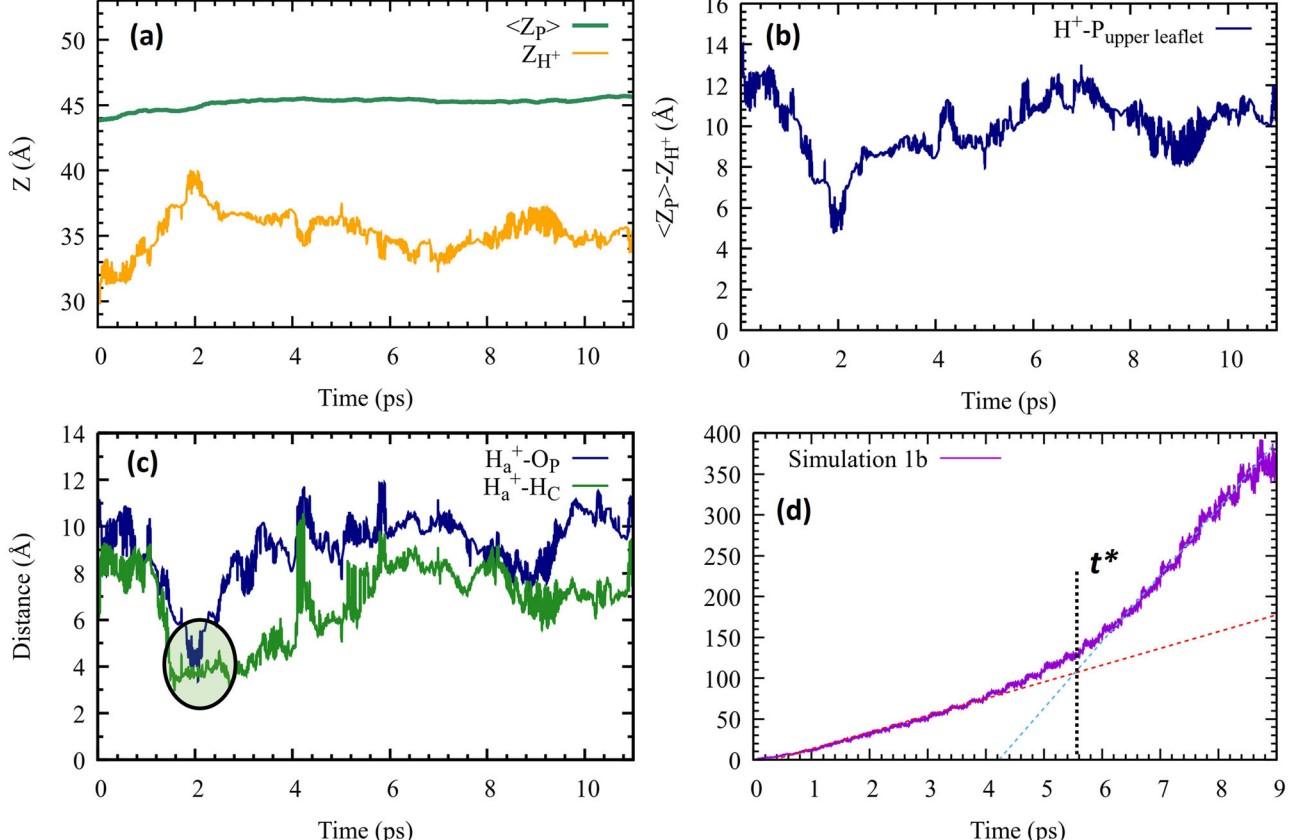

**Fig. 3 | Key insights from Simulation 1b. a** Comparison of the $Z$ coordinate of the excess proton (yellow line, $Z_{H^+}$) with that of the average P-plane in the upper leaflet (green, $<Z_P>$). **b** Their difference, depicting the distance of the plane from the proton indicator. **c** Illustrates the distance of $H_a^+$ from the nearest phosphatic

oxygen ($O_P$; shown in blue) and choline hydrogen ($H_C$; shown in green). **d** Multiple-origin MSD, up to 9 ps. Red and cyan dotted lines depict linear fits to the MSD within the time intervals of 0–4.5 ps and 6–9 ps, respectively. Source data are provided as a Source Data file.

at a distance of about 10 Å from the upper leaflet, until the end of this simulation (Fig. 3b).

For quantifying the proton's diffusion rate, we have calculated the mean squared displacement (MSD) vs. time as depicted in Fig. 3d. To improve the statistical accuracy, we employed here the multiple origin method, while results using a single origin are shown in Supplementary Fig. 1. In this method, multiple trajectory segments are generated from

a single trajectory by truncating progressively longer sections from its origin (the truncated segments are in multiples of $\Delta t$, here $\Delta t = 250$ fs). The new origins are superimposed and the segments averaged together.

We find that the MSD is not linear over the entire timescale. Rather, it changes slope at time $t^*$. Therefore, we fit two linear equations, one around 3 ps and the other around 7 ps (dotted lines in

Fig. 3d). Division by 6 gives two three-dimensional diffusion coefficients, $D_{short}$ and $D_{long}$, as summarized in Table 1. In all cases $D_{long} > D_{short}$ and both are larger than $D_{H^+}$, the proton diffusion coefficient in pure water.

## Multiple protons

We have seen that a single in silico proton at the membrane-water interface showed limited lateral diffusion, with a strong tendency to move toward lipid headgroups. Their collision either led to a covalent PO-H bond formation or to repulsion from the choline group. One possibility that has not yet been tested is the titration of headgroup sites by the first arriving protons, and rapid diffusion of the remainder[17]. We address this lacuna systematically in Simulations 2 and 3.

In Simulation 2a, we have introduced two excess protons: $H_a^+$ near the lower leaflet, and $H_b^+$ farther away from it (see Fig. 1). We found that $H_a^+$ instantly engaged in a covalent bond (length 1 Å) with the nearest $O_P$ atom and remained attached to it throughout the simulation, see Fig. 4a. Its inset shows that the nascent covalent bond had excess

energy, manifested in high-amplitude oscillations, which relaxed in about 200 fs.

In contrast, $H_b^+$ initially migrated towards the bulk, reaching the middle of the water layer, 12.5 Å from the membrane surface (Fig. 4b). It subsequently diffused parallel to the membrane for about 2 ps. It then altered its course, veering towards an $O_P$ atom from the upper leaflet, inducing water-wire formation leading to rapid proton binding (4.2 ps). From this time onwards, both protons were covalently bound to phosphate groups, on opposing leaflets of the membrane (see Supplementary Movie 3.mp4). Figure 4c shows the MSD of $H_b^+$ until 3.5 ps, just before its immobilization on the surface. Again we observe two distinct $D$ values, as reported in Table 1. Interestingly, these values are close to those of $H_a^+$ from Simulation 1b, even though the conditions for the two trajectories were different.

In Simulation 2b, we initiated the system with one excess proton already attached to a phosphate group in the lower leaflet. Over the course of the 11 ps simulation, this covalent bond remained stable, losing some of its excess energy only around 7 ps (Fig. 4d). In contrast, $H_b^+$ exhibited significant lateral mobility (Fig. 4e), see also Supplementary Movie 4.mp4.

The distance expected to be constant (on average) during lateral mobility is the proton's distance to the mean planar interface of the phosphate atoms, $< Z_P >$, which is depicted in Supplementary Fig. 3. The proton indicator distance from this plane is

$$\zeta \equiv Z_{H^+} - <Z_P> \tag{1}$$

This is a better choice than the proton's distance to the nearest $O_P$ atom, used elsewhere[27], as explained in the Coordinates subsection.

Taking the width of each hydration layer as roughly 3.5 Å [the first minimum in the water-water $g(r)$], the second- and third-shells end roughly at $\zeta = 7$ and 10 Å, respectively. The flat segments in $Z_{H^+}$ of Fig. 4e can be interpreted as lateral proton motion in the second or third hydration layers of one or the other leaflets.

**Table 1 | Three-dimensional diffusion coefficients, $D_{short}$ and $D_{long}$ (in Å²/ps), obtained from linear fits to the MSD in the short ($0 \le t < t^*$) and long ($t^* < t$) time ranges, respectively**

| Simulation (proton) | $t^*$, ps | $D_{short}$ | $D_{long}$ |
|---|---|---|---|
| 1b ($H_a^+$) | 5.6 | 3.4 | 13.2 |
| 2a ($H_b^+$) | 2.1 | 3.8 | 10.8 |
| 2b ($H_b^+$) | 5.1 | 1.28 | 2.08 |
| 3a ($H_c^+$) | 6.0 | 1.62 | 6.62 |
| 3b ($H_c^+$) | 4.5 | 1.46 | 2.60 |
| pure water (H⁺) | 10.0 | 0.86 | – |

The result for an excess proton in pure water (Supplementary Fig. 2) is given in the last row. It is close to the experimental value of 0.93 Å²/ps[20], supporting the use of DFTB3 for proton diffusion.

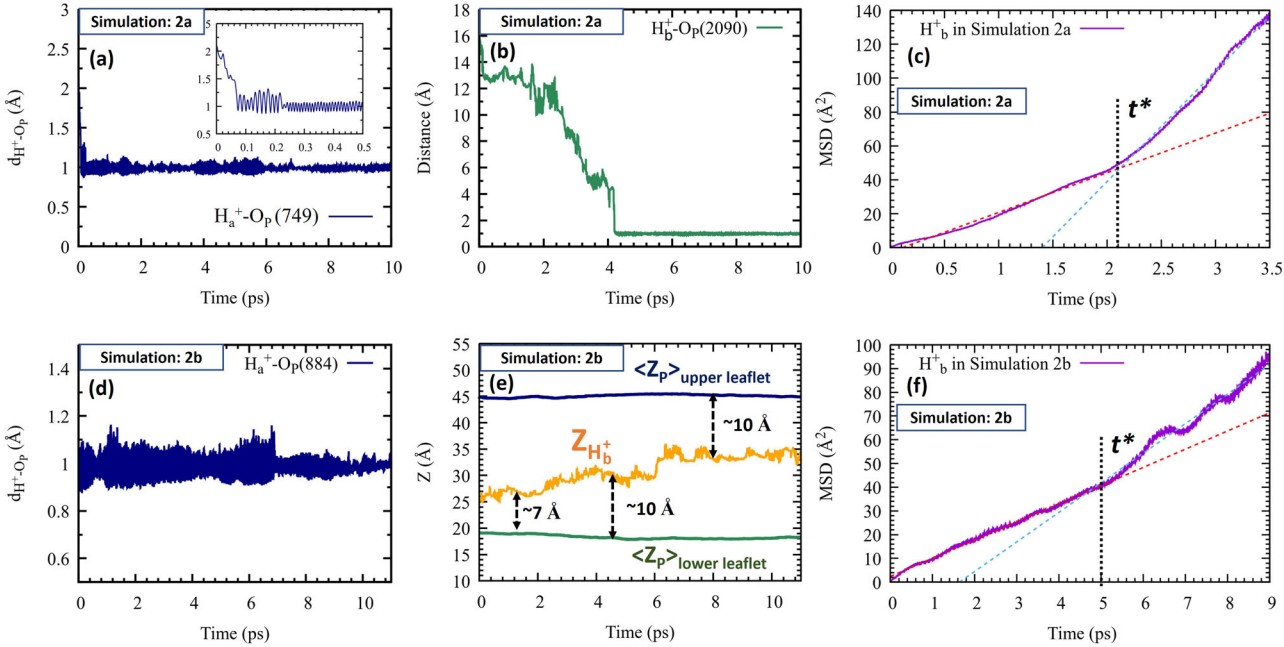

**Fig. 4 | Highlights from Simulation 2.** Simulation 2a (upper panels): Distance of the excess protons (**a**) $H_a^+$, and (**b**) $H_b^+$, from the nearest phosphatic oxygen ($O_P$) on the lower (in blue) and upper (in green) leaflets, respectively. **c** The MSD of $H_b^+$ up to 3.5 ps, with linear fits (red and cyan dashed lines) from which the short- and long-time diffusion coefficients were obtained (see Table 1). Simulation 2b (lower panels): **d** The distance of $H_a^+$ from the $O_P$ it was initially attached to, and (**e**) Comparison of the $Z$ coordinate of the mobile proton ($H_b^+$) (yellow line) with that of the average P-plane in the lower (green) and upper leaflets (blue). **f** The MSD of $H_b^+$ with linear fits (red and cyan dashed lines) from which diffusion coefficients were obtained (Table 1). Source data are provided as a Source Data file.

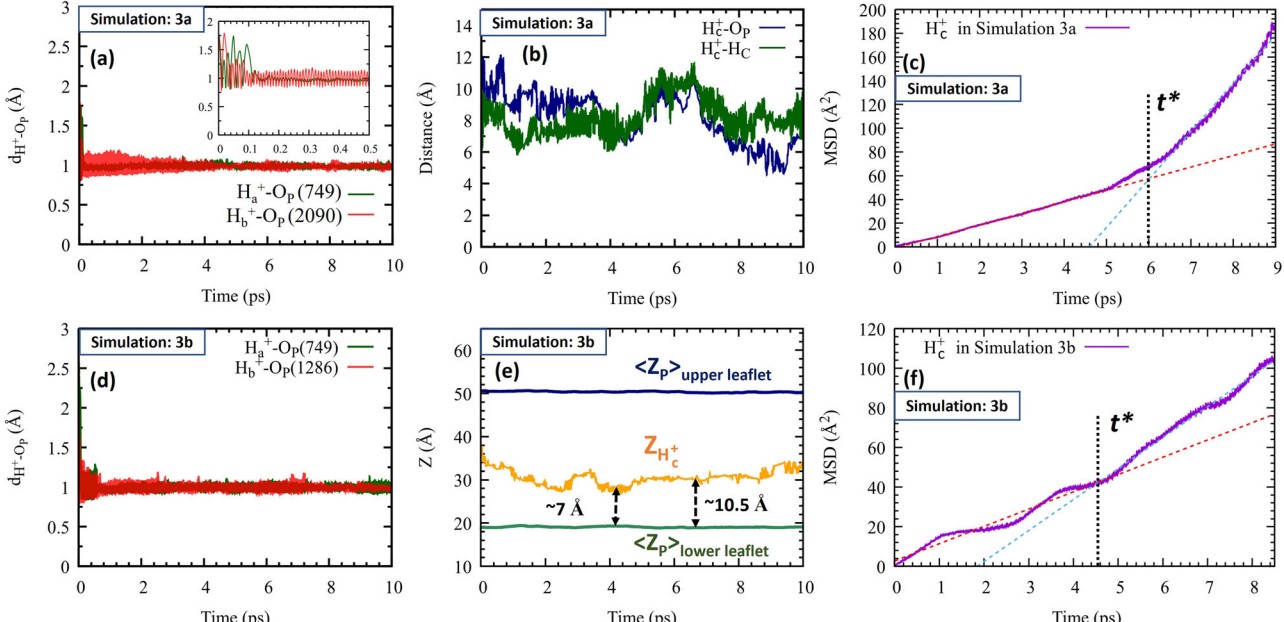

**Fig. 5 | Highlights from Simulation 3.** Simulation 3a (upper panels): (**a**) The distances of $H_a^+$ and $H_b^+$ from the nearest $O_P$, and (**b**) the distance of $H_c^+$ from the nearest $O_P$ (in blue) and the nearest choline hydrogen (in green). **c** The MSD of $H_c^+$ with linear fits (red and cyan dashed lines), from which diffusion coefficients were obtained (Table 1). Simulation 3b (lower panels): (**d**) The distances of $H_a^+$ and $H_b^+$ from the nearest $O_P$, and (**e**) comparison of the $Z$ coordinate of the excess proton ($H_c^+$, yellow line) with that of the average P-plane in the lower (green) and upper leaflets (blue). **f** The MSD of $H_c^+$ with linear fits (red and cyan dashed lines) from which diffusion coefficients were obtained (Table 1). Source data are provided as a Source Data file.

Thus after approximately 2 ps, $H_b^+$ in Simulation 2b migrated from the second solvation layer of the lower leaflet, where it was initially placed, to its third layer, conceivably by utilizing its initial excess energy. This implies that the proton's free energy is lower in the second- vs. third-layer. By 7 ps, it entered the third solvation layer of the upper leaflet (Fig. 4e). By this time it has lost most of its excess energy (manifested by the reduced-amplitude oscillations in Fig. 4d), so that the transition between two 3rd-hydration layers is nearly isoenergetic.

The MSD of $H_b^+$ is shown in Fig. 4f, revealing, again, two distinct diffusion coefficients, as reported in Table 1. The ratio $D_{long}/D_{short}$ is smaller than in the other simulations, perhaps because both diffusion coefficients originate from diffusion in the same hydration layer (of opposing leaflets).

In Simulation 3a we introduced one proton near each leaflet ($H_a^+$ and $H_b^+$), and a third ($H_c^+$) in the bulk (see Fig. 1). The first two promptly formed covalent bonds with their nearest phosphatic groups (Fig. 5a), with most of the excess energy dissipating within ca. 200 fs (inset). In contrast, $H_c^+$ moved laterally during the entire simulation (see also Supplementary Movie 5.mp4), without colliding with the interface (Fig. 5b).

Figure 6 shows an alternate representation of Fig. 5b, now for the $H_c^+$ distance from the upper leaflet phosphate plane. During the first ps, $H_c^+$ possessed excess energy, propelling it to mid-water. Then (1–4 ps) it lost energy, diffusing in the 3rd-hydration layer (10–11 Å). Subsequently (ca. 7–10 ps) $H_c^+$ lost more energy restricting its motion to the 2nd solvation-layer (ca. 7–8 Å). Thus, as noted above, the proton's free energy is probably lowest in the 2nd hydration layer.

Moreover, the transition time from the third to the second layer roughly coincides with $t^* \approx 6$ ps, when the lateral diffusion coefficient transitioned from $D_{slow}$ to $D_{high}$. Presumably, lateral proton diffusion is fastest in the second hydration layer, where water wires are parallel to the interface, so that the increase in $D$ might signal transition to the second layer. This systematic behavior (in comparison to Fig. 5b) is revealed here after abandoning measuring the proton's shortest distance to a P-atom[27] in favor of the shortest distance to the P-plane.

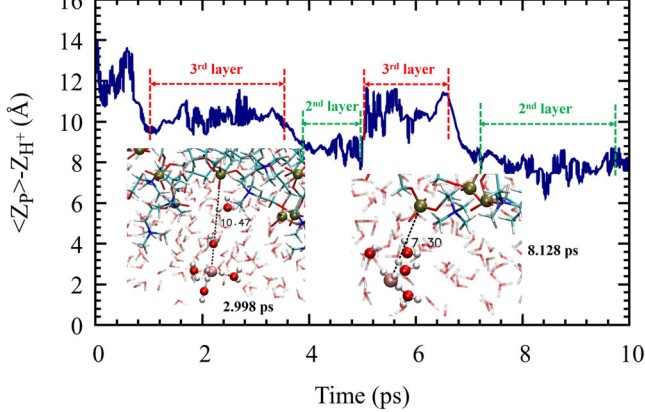

**Fig. 6 | Proton distance from the phosphate plane in Simulation 3a.** Blue line: Distance, along the membrane normal, between the diffusing proton, $H_c^+$, and the average phosphate plane of the upper leaflet. Insets: Snapshots at 3 and 8.1 ps, with dashed lines measuring distances from the proton to its nearest P atom. Source data are provided as a Source Data file.

Simulation 3b is similar to 3a but with an extra water layer. This presents a decisive test for the mobile proton: Would it still prefer travelling near the membrane surface or in mid-water? Fig. 5e shows that the mobile proton, $H_c^+$, initially midway between the two leaflets (at $Z = 35$ Å), approached the lower leaflet within 3 ps. Subsequently, it moved parallel to the membrane surface, through its second and third hydration layers, but never in mid-water (see also Supplementary Movie 6.mp4).

The lateral motion of the proton indicator in four of our simulations is shown in Fig. 7. During the first 2 ps, it undergoes large excursions (purple lines), attributed to its excess energy. This agrees with Supplementary Fig. 4b showing that most of the potential energy relaxation occurs in the first 2 ps. For longer times, the proton in

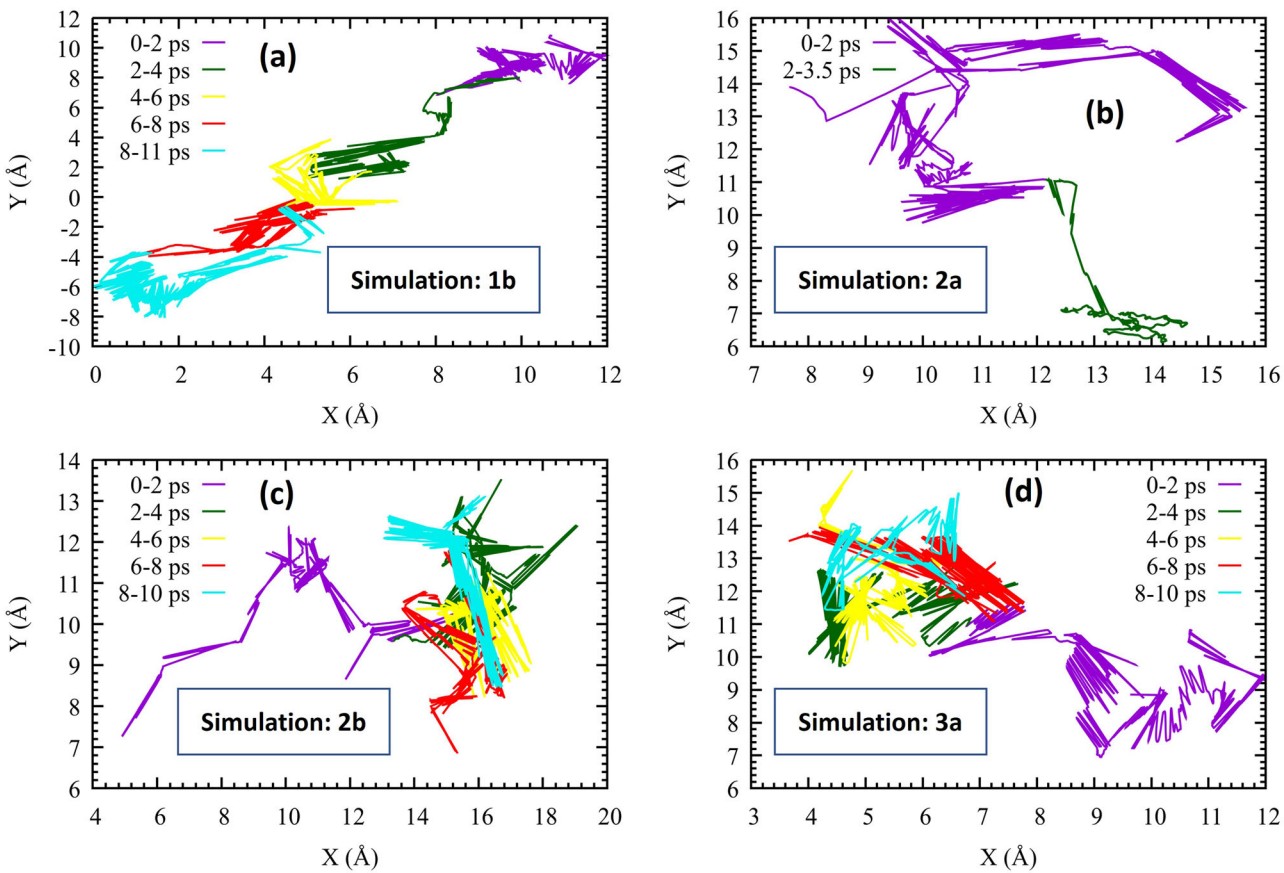

**Fig. 7 | Lateral proton dynamics.** Depiction of unbound proton excursions in the *XY* plane (irrespective of *Z*) for (**a**) Simulation 1b, (**b**) Simulation 2a, (**c**) Simulation 2b, and (**d**) Simulation 3a. Each trajectory is segmented into 2 ps intervals, represented with distinct colors. Source data are provided as a Source Data file.

Simulation 1b shows large excursions (commensurate with its large *D* value), whereas Simulations 2b and 3a undergo compact explorations, with small *D* values.

### Sodium simulation

For comparison, we simulated an additional system containing $Na^+$ ions (8 + 8 POPC lipids, 477 water molecules, and 3 NaCl) at 300 K. We find, as in our earlier work on ions at the water-lipid interface[30], that while some $Na^+$ cations interact with the phosphatic headgroups (no covalent bonds), the remainder diffuse freely in the water phase (see Supplementary Movie 7.mp4). However, their diffusion coefficient, $D \approx 0.06$ Å²/ps (Supplementary Fig. 5), is notably smaller than 0.096 Å²/ps found in our previous work for $Na^+$ in aqueous solution of 400 mM NaCl (Fig. S23f in ref. [30]). The latter is very close to tracer diffusion experiments for 0.4 M NaCl at 298 K, $D = 0.116$ Å²/ps (Eq. 5a in ref. [31]). Hence, unlike the proton, sodium diffusion in the membrane hydration water is slower than in pure water. Moreover, its MSD (Supplementary Fig. 5) is linear over the whole time regime, without manifesting the two time-scales found here for protons.

### Ab initio MD simulations

Our 2.8 ps BOMD Simulation 2a′ provides supporting evidence for phosphate group protonation (in contrast to CPMD results showing only hydrogen-bond formation)[27]. As in Simulation 2a, we have introduced two protons: One near the membrane ($H_a^+$) and the other away from it ($H_b^+$). Their distances from the nearest phosphatic oxygen are illustrated in Supplementary Figs. 6a,b. Similarly to Simulation 2a, $H_a^+$ instantaneously protonated $O_P$, which remained protonated up to 2.8 ps (Supplementary Fig. 6a). After 0.5 ps, $H_b^+$ approached the membrane, protonating another $O_P$ (Supplementary Fig. 6b). However,

unlike $H_b^+$ of Simulation 2a (Fig. 4b), it did not bind continuously but rather fluctuated between the bound state and the two hydration shells of $O_P$, indicating that binding two protons on the same leaflet is unfavorable.

## Discussion

Over sixty years have passed since Mitchell's chemiosmotic theory, and the fundamental question of how/whether protons move near biological membranes remains unanswered. As detailed in the Introduction, experiments from the Pohl group showing that protons move rapidly along a membrane's surface without dissipating into the bulk remain controversial. One troubling aspect is the discrepancy with molecular simulations. Various flavors of quantum-like MD were recently utilized, agreeing that a hydronium ion in the water-membrane interface will rapidly diffuse towards a phosphatic headgroup, forming a strong hydrogen-bond that prevents further proton motion, certainly not as rapid as in liquid water.

Here we used the DFTB3 methodology to substantially speed up calculations of single- and multiple-protons near a POPC membrane, with BOMD and $Na^+$ trajectories as control. In agreement with the previous simulations, we found that a single excess proton near a membrane will rapidly diffuse towards the phosphate headgroup by utilizing a short water wire of ca. 3 water molecules. However, the formation of a PO−$H^+$ covalent bond is observed here for the first time. When the hydronium is initially in the midst of the water pool, its path toward the phosphate may be obstructed by the choline group, repelling it back into solution.

One or two protons, initially near the upper or lower leaflets, instantaneously protonate them for the duration of the simulation. An additional proton, being repelled from both leaflets, diffuses laterally in

the bulk water pool, in line with the "proton front" scenario of Agmon and Gutman[17]. Such protons, observed in Simulations 2b, 3a, and 3b, are found in the 2nd- or 3rd-hydration layers (except during the first 2 ps, when they still possess excess energy). To verify this, we have included an additional water layer in Simulation 3b, so that the middle of the water pool is the 4th layer with respect to either leaflet. The mobile proton did not move there, but rather stayed approximately 7–10 Å from the membrane surface, commensurate with experiments[16,19].

Unlike sodium, the proton appears to diffuse faster at the interface than in pure bulk water[17], and with two different diffusion coefficients: One up to $t^* \approx 6$ ps, and an even larger $D$ thereafter (Table 1). For a single proton (Simulation 1b), attraction to surface phosphate groups could be speeding the proton mobility. For the multiple proton simulations 2b, 3a and 3b in which lateral mobility occurs, $D_{short}$ is similar, and just slightly larger than in bulk water. For $t > t^*$ in Simulation 3a, the mobile proton diffuses in the 2nd-layer and $D_{long}$ is excessively large. Possibly, lateral diffusion is fastest in the 2nd-layer. Whether this could be ascribed to water-wires parallel to the interface, remains to be investigated.

## Methods

### Classical MD

Excepting Simulation 3b, the system consisted of 16 palmitoyl-oleoyl-phosphatidylcholine (POPC) lipids (134 atoms per lipid), 8 lipids in each leaflet, along with 458 water molecules, resulting in a total of 3518 atoms. The system was placed in a rectangular box with edge lengths of 22.4 × 22.4 × 65.2 Å. The long axis is defined as the $Z$ axis. Out of its ca. 65 Å length, the membrane thickness was approximately 40 Å, and hence the water pool was 25 Å thick. Its volume was thus $V_W = 12, 550$ Å$^3$. The "concentration" of a single proton, $1/(V_W N_A) = 0.13$ M, corresponds to pH $\approx 1$, $N_A$ being Avogadro's number. This is close to the pH of the solution in the micropipette droplet used experimentally[26].

In Simulation 3b, the water pool was expanded by one additional layer, resulting in 554 water molecules (vs. 458 above) within a 22.4 by 22.4 by 68.4 Å rectangular box. Thus, the thickness of the water pool has increased to ca. 28.5 Å. Assuming a water layer thickness of 3.5 Å, there are 8 water layers separating the two leaflets here, as opposed to 7 layers in all other DFTB3 simulations.

The initial configuration of the system was generated using the CHARMM-GUI web-based graphical interface[32]. The lipids were treated using the Lipid21 force field as implemented in AMBER[33], while the improved four-site TIP4P-Ew water model[34] was used for water.

Classical all-atom MD simulations for this system were performed within the framework of the Gromacs 2020 package[35]. The simulation protocol comprised of an initial energy minimization for 500 ps, followed by equilibration under the NPT ensemble (constant number of particles, pressure and temperature) for a total of 2 ns. The equilibration was performed in six consecutive steps by gradually relaxing the positional and dihedral restraints for the lipids (details in Supplementary Table 1). During the simulation, all covalent bonds involving hydrogen atoms were constrained using the Linear Constraint Solver algorithm[36]. Periodic boundary conditions (PBC) were imposed on all three cartesian coordinates. The particle-mesh Ewald summation method[37] with a cutoff distance of 9 Å, was used to evaluate the long-range electrostatic interactions emanating from the PBC.

Equilibration was continued for an additional 20 ns, with a time step of 2 fs under the NPT ensemble, for a temperature of 300 K and a pressure of 1 atm. Temperature was controlled using Langevin dynamics with a damping coefficient of 1 ps$^{-1}$, while the pressure was controlled using the Langevin piston Nosé-Hoover method[38].

### DFTB3 simulation of proton transfer

The transfer of protons in chemical or biological systems involves cleavage and formation of covalent bonds, necessitating a quantum

mechanics (QM)-based approach for the electronic structure. However, employing high-level QM methods involves computational challenges, such as severe limitations on system size and simulation times. One approach for overcoming these obstacles applies mixed quantum-mechanics/molecular-mechanics (QM/MM) algorithms, which reduce the size of the quantum subsystem. However, for a rapidly diffusing proton the QM subsystem has to be redefined every few timesteps. This "adaptive QM/MM" has been implemented for proton in bulk water[39], or in specific biological channels[40], but not for more complex protonated systems. Alternately, approximate QM methods have proven to be valuable, as they allow access to larger systems and longer times, while maintaining a good compromise between accuracy and computational cost.

One such method is Self-Consistent-Charge Density-Functional Tight-Binding (SCC-DFTB)[28]. Inclusion of the third-order energy correction (DFTB3)[41] has substantially improved the description of charged systems (e.g., zwitterionic membranes) containing the elements C, H, N, O, and P. Here we used a simpler DFTB version that requires calculating only the diagonal terms in the 3rd-order expansion (DFTB3-diag), which is faster to compute, and also the only DFTB3 version currently implemented in the CP2K package[42].

DFTB3-diag with the MIO parameter set was tested for the structure of water in Fig. 6a of ref. 43, which was repeated herein (Supplementary Fig. 7). While DFTB3 captures the first hydration shell of water reasonably well, it shows limited accurately in describing the higher hydration shells. Full DFTB3 with the 3OBw parameter set shows notable improvements in depicting water structure, but no improvements in water dynamics, as manifested in the water self-diffusion coefficient[44]. It is also computationally costly, and beyond practicability for full QM calculations.

Our DFTB3 simulation was initiated from the last snapshot of the classical MD simulation (Sec. 2), and propagated for 2 ps at 300 K in the NVT ensemble for further equilibration. Temperature was controlled by a Nosé-Hoover thermostat. The kinetic and potential energies were monitored during the simulation, see Supplementary Fig. 4. Their time evolution becomes nearly constant within 1 ps, when the system has apparently reached equilibrium.

To introduce excess protons, we used the final DFTB3-equilibrated geometry and replaced one or more water molecules with gas-phase-optimized hydronium ions ($H_3O^+$). The position of the oxygen atom was kept unchanged during this process, which was performed using the GaussView 6 software[45]. The DFTB3-diag NVT production runs for the protonated system utilized the DFTB3 Mio-1-1 parameter set with the D3 dispersion correction. Ewald-type methods were applied to account for Coulomb interactions. Other parameters were set to the CP2K default values.

As a reference, we also simulated pure water with one excess proton and without the membrane in a 25.47 Å cubic box.

### Ab initio MD

To further assess the accuracy of the DFTB3 method, we have also conducted one BOMD simulation under identical conditions to our DFTB3 Simulation 2a (hereafter Simulation 2a′). It was propagated for a shorter duration of 2.8 ps due to its high computational cost.

For BOMD simulations, we employed the QUICKSTEP module[46] within the CP2K package[42]. Forces were computed using density functional theory (DFT), with the Becke-Lee-Yang-Parr (BLYP)[47,48] functional, and the added empirical dispersion correction term (BLYP-D3)[49]. We have used the double-zeta valence polarization basis set (DZVP-MOLOPT-SR)[50], augmented with the Goedecker-Teter-Hutter (GTH) pseudo-potentials[51,52]. A grid level cutoff of 70 Ry was applied to the Gaussian basis set, while the plane-wave basis function employed a cutoff of 320 Ry.

## Proton indicator

While the dominant proton transport mechanism in water is the stepwise Grotthuss mechanism[20], where the proton hops from a distorted hydronium to its neighboring water molecule, there is a notable contribution also from a concerted mechanism of protons hopping along a hydrogen-bonded water chain ("water wire")[21,53]. This mechanism was also identified at the membrane-water interface[27], particularly near a phosphate group.

For analyzing stepwise proton transport in aqueous environments, Lin and collaborators have developed a "proton indicator" algorithm[54], which captures smoothly the average location of the proton as a function of time. Extending this to concerted mechanisms, Lin et al.[29] have recently developed an enhanced indicator mechanism ("Indicator 2"), which we utilized here to report the proton's distance from key lipid atoms (the phosphatic oxygen, $O_P$, and cholinic hydrogen, $H_C$).

## Coordinates

When a proton reacts with a phosphatic oxygen, $O_P$, it is natural to use the $H^+ - O_P$ separation as the proton coordinate. However, our interest is not limited to pair interactions, because the proton might diffuse laterally over long distances. As a generalization, Nguyen et al.[27] (see their Fig. 5) suggested using the minimal $H^+ - O_P$ distance, $d_{min}(H^+ - O_P)$. This is a discretized, inaccurate measure of the distance between a point (the diffusing proton) and a plane (the interface):

a. The identity of the closest $O_P$ changes abruptly with time, and so does $d_{min}(H^+ - O_P)$;
b. Any single $O_P$ atom fluctuates randomly perpendicular to the plane, see Supplementary Fig. 3, and these fluctuations will be added to the proton fluctuations.

As an alternative, we have used the proton's $Z$-coordinate, $Z_{H^+}$, which the program calculates every timestep. We averaged $Z$ over the P-atoms of a given leaflet, $< Z_P >$, and calculated the difference $|Z_{H^+} - <Z_P>|$. This is rigorously the proton distance from the P-atom plane that does not include fluctuations from individual P-atoms.

## Reporting summary

Further information on research design is available in the Nature Portfolio Reporting Summary linked to this article.

# Data availability

The data generated in this study for all XY plots in all figures are available in the Source Data file. The initial and final coordinates of all DFTB3 trajectories are included in the Supplementary Data 1 file. Truncated versions of all trajectories are provided as seven movie files within the Supporting Information. Source data are provided with this paper.

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

## Acknowledgements
This research was supported by the Israel Science Foundation grants Nos. 722/19 N.A. and 817/24 N.A.

## Author contributions
S.M. performed the simulations, prepared the figures and wrote the first version of the manuscript. N.A. designed the research, suggested interpretations and wrote the final version.

## Competing interests
The authors declare no competing interests.
