## [Transparent Peer Review file · Nature Communications]

Multi-Proton Dynamics near Membrane-Water Interface

Corresponding Author: Professor Noam Agmon

Version 0:

Reviewer comments:

Reviewer #1

(Remarks to the Author)

This manuscript reports an interesting computational study of proton dynamics near membrane lipid bilayers. This work aims to pinpoint the discrepancy between previous experimental and computational findings regarding the protonic electrochemical potential gradient. More specifically, previous experiments revealed rapid proton motion over large distances at the membrane-water interface, while previous computational showed an excess proton being immobilized by a lipid headgroups. This is an important subject of great interest to a wide audience.

The authors compared a number of models with various number of excess protons. They found that, while some protons were indeed trapped by the lipid headgroups, other protons could diffuse laterally more quickly than in bulk water. The authors thus suggested that their findings rationalize the experimental observations.

Overall, the manuscript is clearly written and easy to follow. The methods chosen are reasonable. The author attempted to validate the DFTB3 model by additional short DFT simulations. The major conclusions agree with our physical intuition.

My major concern is that the limitations of the present study were not sufficiently scrutinized or discussed. The authors may want to examine the possible pitfalls of the employed methods, small size of the models, limited number (only 1) of trajectories for each scenario, and short simulation times. It is understandable that the simulations are expensive and that we have to live with certain limitations, but a detailed analysis on these weaknesses would improve the manuscript. It would also be better to emphasize more on the overall qualitative picture than on the specific quantitative results.

Minor points:

1. The "multi-origin method" for estimating the proton diffusion coefficient should be briefly explained, probably in the supporting information, because not everyone is familiar with this method. Comparing the numerical results with the "single-origin" method in the supporting information will be informative, although not mandatory.
2. Pages 16-17: Which "solvation shells" are discussed here, and how are they quantified?

Reviewer #2

(Remarks to the Author)

In this interesting contribution, the authors employed MD simulations with a semi-empirical DFT method (DFTB3) to probe the behaviors of excess protons near a lipid bilayer. The key difference compared to previous computational efforts along this line include: (1) the higher computational efficiency enabled longer sampling; (2) they included multiple excess protons near the interface. These simulations suggest that the "first" sets of excess protons can form strong interactions with the phosphate group, including (at least) transient covalent bond formation, and therefore remain bound to the membrane surface with a low level of mobility. This is consistent with observations from previous efforts using, for example, MS-EVB models. However, an additional (e.g., the "third") proton is not strongly bound and seems to partition favorably to the second solvation shell of the phosphate and exhibits rapid (faster than bulk) lateral diffusion, in agreement with previous experimental measurements and a theoretical model pioneered by the senior author. Therefore, these new simulations are able to reconcile seemingly contradictory observations from previous experimental and computational studies of proton diffusion near the lipid membrane, which is a topic of major importance in bioenergetics. Therefore, I support the publication

of the work.

I have several questions/suggestions:

1. The MSD analyses revealed two time scales with rather different diffusion constants for the excess proton(s). What's the physical/molecular origin for these two regimes? Interestingly, even the lower diffusion constant is higher than that in bulk water (how does the computed bulk value compare to experiment?). What is the physical origin for such acceleration? Is this because the interfacial water molecules have preferential orientations that somehow facilitate lateral diffusion of the hydronium? Some additional analysis would be informative.
2. I certainly appreciate the computational cost of these all QM (rather than QM/MM) calculations, even at the DFTB3 level. Nevertheless, having only one trajectory for each set up leaves quite a bit of uncertainty. It seems important to run multiple independent trajectories and see if the observations are reproducible. Is this possible at all with the computational resources available to the group? Also along this line, the level of details for the DFTB3 simulations is rather lacking. For example, what program was used to conduct these simulations? What was the thermostat algorithm used?

Reviewer #3

(Remarks to the Author)

The authors have applied both DFTB and DFTMD simulations to investigate the role of multiple excess protons on long-distance proton transport near the membrane-water interface. The manuscript provides a very nice summary of past efforts to solve this mystery from both experiments and simulations. The results are interesting. The following points should be addressed:

The two key questions to this conundrum are a): Why are protons localized at the membrane-water interface? b): How can they move fast(er) near the interface?

1) Regarding the question a), there are extrinsic and intrinsic reasons. Excess proton in the form of hydronium is an amphibolic ion, therefore they will prefer to stay at the water interface. This has been pointed out before, see Ref.24 and <https://doi.org/10.1002/anie.201707391>. The extrinsic reason has to do with the pKa of the lipid group. Given their pKa value is about 6, excess proton (in acidic solution) will protonate these groups. The authors need to point out both factors in their Introduction.

2) Regarding question b), there are also extrinsic and intrinsic reasons. Besides the structural diffusion of protons, Interfacial water has a different lateral diffusion pattern as compared to bulk water. How much does the observed increment in the surface proton diffusion depend on the interfacial water diffusion? This may be useful to use Na⁺ ion as a control to separate these effects.

3) How good is the version of DFTB3 that the authors used to describe the water structure in the bulk? I remember that the DFTB community has been struggling to get the bulk water right. I suggest the authors provide this information (e.g. g_{OO} of bulk water) in their SI. If the method can not do the bulk water properly, then it is more difficult to trust the results of the interface.

Reviewer #4

(Remarks to the Author)

The current study explores multi-proton dynamics to address the discrepancy between the experimentally observed rapid lateral proton mobility and its immobilization at an interfacial phosphate in MD simulations. Using Self-Consistent-Charge Density-Functional Tight-Binding with third-order energy correction (DFTB3), the authors simulate a membrane patch with one, two, or three protons. They systematically demonstrate how the trajectory of one proton significantly influences the others, effectively reconciling experimental and theoretical inconsistencies.

This is a compelling study, and I believe readers of Nature Communications will find it valuable. I have one minor comment: the pH conditions in the study seem quite acidic. How do these conditions compare to existing in vitro studies? Additionally, is it feasible to use bulk pH for comparison, considering that protons are released from the membrane and may not equilibrate with the bulk?

Version 1:

Reviewer comments:

Reviewer #1

(Remarks to the Author)

The authors have addressed my concerns by performing and discussing additional simulations and by including additional

technical details. I am happy to recommend the publication of the manuscript.

Reviewer #2

(Remarks to the Author)

The authors have conducted additional calculations to boost statistics. They have also revised the ms to adequately address the questions I raised in the last round of review.

Reviewer #3

(Remarks to the Author)

My issues have been properly addressed. The paper could now be published.

Reviewer #4

(Remarks to the Author)

The authors addressed my concerns. I have no further comments.

NCOMMS-24-48607, Revision

Mallick and Agmon, "Multi-Proton Dynamics near Membrane-Water Interface"

General:

List of simulations added:

Simulation 2b, Simulation 3b

List of subsections added:

- 1) Result/Initial condition
- 2) Result/Sodium simulation

List of figures added:

- 1) Figures 4d, 4e, and 4f in the main MS for simulation 2b
- 2) Figures 5d, 5e, and 5f in the main MS for simulation 3b
- 3) Figure 7 in the main MS
- 4) Figure S5 in the ESI
- 5) Figure S7 in the ESI

List of equations added:

- 1) Equation 1 on page 12

List of references added:

- 1) Refs. 24, 25, 29, 30, 31, 39, 40, 45

Response to REVIEWER COMMENTS

Our response in Blue

Reviewer #1 (Remarks to the Author):

This manuscript reports an interesting computational study of proton dynamics near membrane lipid bilayers. This work aims to pinpoint the discrepancy between previous experimental and computational findings regarding the protonic electrochemical potential gradient. More specifically, previous experiments revealed rapid proton motion over large distances at the membrane-water interface, while previous computations showed an excess proton being immobilized by lipid headgroups. This is an important subject of great interest to a wide audience.

The authors compared a number of models with various number of excess protons. They found that, while some protons were indeed trapped by the lipid headgroups, other protons could diffuse laterally more quickly than in bulk water. The authors thus suggested that their findings rationalize the experimental observations.

Overall, the manuscript is clearly written and easy to follow. The methods chosen are reasonable. The author attempted to validate the DFTB3 model by additional short DFT simulations. The major conclusions agree with our physical intuition.

Comment 1: My major concern is that the limitations of the present study were not sufficiently scrutinized or discussed. The authors may want to examine the possible pitfalls of the employed methods, the small size of the models, the limited number (only 1) of trajectories for each scenario, and short simulation times.

Response 1: (a) We have included two new simulations in the revised manuscript. Thus, we now have two trajectories (rather than 1) for each number of protons (1, 2, and 3).

(b) Although not specifically mentioned by the Referees, another possible pitfall due to the small system size is that the water pool between the two leaflets is small, so that its midpoint could coincide with the 2nd or 3rd hydration layers. In the new simulation 3b we have expanded the width of the water pool to check whether the mobile proton still prefers the 2nd hydration shell over the middle of the water pool. Indeed, this was now verified (see added Fig. 5d,e,f and its discussion on p. 14).

(c) Concerning the distance and time-scale limitations of previous simulations in the literature we said:

“Due to the heavy demands in computer time for CPMD, these simulations²⁷ were confined to a small membrane patch (up to 10 lipid residues), and 10 ps duration.” (p. 4-5)

The expanded Methods/DFTB3 subsec now explains how by a judicious choice of computational methods we could run a larger model than previously achieved.

Comment 2: (a) It is understandable that the simulations are expensive and that we have to live with certain limitations, but a detailed analysis on these weaknesses would improve the manuscript. (b) It would also be better to emphasize more on the overall qualitative picture than on the specific quantitative results.

Response 2: (a) Indeed, these simulations are computationally demanding, thus we are restricted to just a few trajectories. However, we have now generated two independent trajectories for each case, see 1(a) above. Also, in the DFTB3 subsec. we now give more detail that explains how a judicious balance between accuracy and speed were obtained (p. 19 and 20).

(b) Focusing on the overall qualitative picture, as suggested, we have

b1) Deleted the last column in Tbl. 1.

b2) Added Fig. 7, which gives a qualitative/intuitive view of the lateral diffusion process.

b3) An interesting qualitative interpretation added herein is that the distance of the mobile proton from the interface could be coupled to its excess energy, and that the notable increase of the proton diffusion coefficient at the longer times could be due to proton relaxation into the 2nd hydration layer (see, e.g. the last parag of the Discussion, p. 17-18).

Minor points:

Comment 3: The “multi-origin method” for estimating the proton diffusion coefficient should be briefly explained, probably in the supporting information, because not everyone is familiar with this method. Comparing the numerical results with the “single-origin” method in the supporting information will be informative, although not mandatory.

Response 3:

a) The referee may have overlooked the single-origin MSDs that are shown in Fig. S1 (previously S2) of the supporting info.

b) In the Results sec., after “we employed the multiple origin method” we have added:

“In this method, multiple trajectory segments are generated from a single long trajectory by truncating progressively longer sections from its origin ($\Delta t = 250$ fs). The new origins are superimposed and the segments averaged together” (p. 10).

Comment 4: Pages 16-17: Which “solvation shells” are discussed here, and how are they quantified?

Response 4: The parag following (new) Eq. (1) explains the definition of hydration layers (shells) used herein (p. 12).

Reviewer #2 (Remarks to the Author):

In this interesting contribution, the authors employed MD simulations with a semi-empirical DFT method (DFTB3) to probe the behaviors of excess protons near a lipid bilayer. The key differences compared to previous computational efforts along this line include: (1) the higher computational efficiency enabled longer sampling; (2) they included multiple excess protons near the interface. These simulations suggest that the “first” sets of excess protons can form strong interactions with the phosphate group, including (at least) transient covalent bond formation, and therefore remain bound to the membrane surface with a low level of mobility. This is consistent with observations from previous efforts using, for example, MS-EVB models. However, an additional (e.g., the “third”) proton is not strongly bound and seems to partition favourably to the second solvation shell of the phosphate and exhibits rapid (faster than bulk) lateral diffusion, in agreement with previous experimental measurements and a theoretical model pioneered by the senior author. Therefore, these new simulations are able to reconcile seemingly contradictory observations from previous experimental and computational studies of proton diffusion near the lipid membrane, which is a topic of major importance in bioenergetics. Therefore, I support the publication of the work.

I have several questions/suggestions:

Comment 5: The MSD analyses revealed two time scales with rather different diffusion constants for the excess proton(s). What’s the physical/molecular origin for these two regimes? Interestingly, even the lower diffusion constant is higher than that in bulk water (how does the computed bulk value compare to experiment?). What is the physical origin for such acceleration? Is this because the

interfacial water molecules have preferential orientations that somehow facilitate lateral diffusion of the hydronium? Some additional analysis would be informative.

Response 5: Indeed, a tentative answer to this puzzle is included in the revised ms.

First, we now make the following two of our observations:

(a) The proton is more stable in the 2nd-hydration layer

(b) The proton initially possesses extra energy (by construction). Thus, as this energy dissipates the proton tends to move into the 2nd shell.

Consequently, if lateral diffusion is faster in the 2nd-shell (e.g., due to hydrogen-bond networks parallel to the membrane), proton diffusion will speed up with time.

Discussion of these issues was added on p. 14. In particular, the 2nd parag thereof suggests a tentative explanation for the acceleration issue. We return to these questions in the closing parag of the Discussion (p. 18).

Additionally, we have added to Tbl. 1 the experimental proton diffusion in pure water in its legend: "It is close to the experimental value of 0.93 Å²/ps, supporting the use of DFTB3 for proton diffusion."

Comment 6: I certainly appreciate the computational cost of these all QM (rather than QM/MM) calculations, even at the DFTB3 level. Nevertheless, having only one trajectory for each set up leaves quite a bit of uncertainty. It seems important to run multiple independent trajectories and see if the observations are reproducible. Is this possible at all with the computational resources available to the group?

Response 6: In the revision, we have generated two new trajectories, culminating in *two* independent trajectories for each number of protons (1, 2, and 3), as summarized in (revised) Fig. 1. See also Response 1(a). Consequently, the notations have now been changed to Simulations 1a and b (single proton), Simulations 2a and 2b (2 protons), and Simulations 3a and 3b (3 protons). Incidentally, in the DFTB3 Subsec. we have added a parag. explaining why we have not used a QM/MM approach herein (p. 19).

Comment 7: Also along this line, the level of details for the DFTB3 simulations is rather lacking. For example, what program was used to conduct these simulations? What was the thermostat algorithm used?

Response 7: The DFTB3 Subsec. has been substantially expanded to provide the missing detail, and justify the exact choice of computational methods (pp. 19-20).

(a) We explain why we did not apply QM/MM here, and therefore require the most efficient QM algorithm, DFTB3-diag.

(b) We have added detail, such as the program and parameter set utilized, the thermostat algorithm and more.

Reviewer #3 (Remarks to the Author):

The authors have applied both DFTB and DFTMD simulations to investigate the role of multiple excess protons on long-distance proton transport near the membrane-water interface. The manuscript

provides a very nice summary of past efforts to solve this mystery from both experiments and simulations. The results are interesting. The following points should be addressed:

The two key questions to this conundrum are a): Why are protons localized at the membrane-water interface? b): How can they move fast(er) near the interface?

Comment 8: Regarding the question a), there are extrinsic and intrinsic reasons. Excess proton in the form of hydronium is an amphibolic ion, therefore they will prefer to stay at the water interface. This has been pointed out before, see Ref.24 and <https://doi.org/10.1002/anie.201707391>. The extrinsic reason has to do with the pKa of the lipid group. Given their pKa value is about 6, excess proton (in acidic solution) will protonate these groups. The authors need to point out both factors in their Introduction.

Response 8: First, we note that water-air vs water-lipid interfaces are different, as is now discussed in the Introduction (p. 4):

“The MS-EVB simulations²³ show that one or more of the hydronium’s (Eigen or Zundel forms) OH moieties are hydrogen-bonded to the membrane’s PO or CO groups, hence face away from the water phase. This is quite the opposite compared to the hydronium at the air-water interface, where the hydrogens point towards the water phase.²⁴ Moreover, in the latter case the proton attraction to the interface is weak, $1 k_B T$ (experimentally 1.3 kcal/mol)²⁵, whereas the binding to the membrane according to the MS-EVB simulation amounts to 5 kcal/mol. The common denominator between these two fundamental examples for protons at aqueous interfaces is an orientation that maximizes the hydrogen-bonding interactions.”

Note also that the added ref. 24 here is the reference pointed out in Comment 8 above.

Concerning protonation, we do indeed show that some of the protons protonate the lipid headgroups, but these become immobile protons (at least during our short simulation times). The question we pose is why do the *mobile* protons diffuse close to the interface, rather than in bulk water. Our simulations are indicative of confinement to the 2nd- or 3rd-hydration layer, where the mobile proton has lower free energy.

We added these points in our revised MS:

- a) Given the two added simulations, the immobile protons that protonate the lipid headgroups by forming a covalent bond are observed in nearly all of the simulations: Simulation 1a (Fig. 2a p. 8), H_a^+ in Simulations 2a and 2b (Figs. 4a 4d; p. 12), H_a^+ and H_b^+ in Simulations 3a and 3b (Figs. 5a 5d; p. 13). It is also observed in our AIMD simulation.
- b) Evidence for lower protonic free energy in the 2nd-hydration shell is now discussed in the last parag of p. 12 and the first parag of p. 14.
- c) Most significant is the added simulation 3b (Fig. 5e), with the extended water pool, showing that the mobile proton prefers to diffuse in the 2nd layer rather than midway between the two leaflets (added on p. 14).

Comment 9: Regarding question b), there are also extrinsic and intrinsic reasons. Besides the structural diffusion of protons, Interfacial water has a different lateral diffusion pattern as compared to bulk water. How much does the observed increment in the surface proton diffusion depend on the interfacial water diffusion? This may be useful to use Na⁺ ion as a control to separate these effects.

Response 9: We show that there is indeed a correlation between the location of the proton and its diffusion rate, which is largest in the 2nd-hydration layer (p. 17-18), e.g. last 2 lines in the Discussion (p. 18). At this stage, we can only speculate that this may be due to the hydrogen-bonding pattern in that layer.

Concerning interfacial water diffusion, we have not studied it in this work, but it is expected to be much slower than interfacial proton diffusion at the membrane-water interface. However, Laage *et al* (<https://doi.org/10.21203/rs.3.rs-5099492/v1>) claim that, at the air-water interface, water diffusion speeds up while proton diffusion slows down, in which case interfacial proton and water diffusion occur at similar rates.

We appreciate the referee's suggestion to repeat the simulation(s) with sodium in place of hydronium.

We have done so for the initial conditions of Simulation 3a, with results in the added Sec. "Sodium simulations" (p. 16), and the MSD of Figure S5. In contrast to H⁺, Na⁺ diffuses at the interface slower than in the bulk and does not exhibit dual diffusion rates.

Comment 10: How good is the version of DFTB3 that the authors used to describe the water structure in the bulk? I remember that the DFTB community has been struggling to get the bulk water right. I suggest the authors provide this information (e.g. $g_{\text{O-O}}$ of bulk water) in their SI. If the method can not do the bulk water properly, then it is more difficult to trust the results of the interface.

Response 10: We have added this discussion in the last full parag. on p. 20 (starting with DFTB3-diag). The DFTB3-diag $g_{\text{O-O}}(r)$ plot is added as Fig. S7, showing similar deficiencies as in Fig. 6a of ref. 43. Briefly, DFTB3-full (with the 3OBw parameter set) fixes the disagreement however (a) is much more expensive computationally and (b) does not improve on the diffusion coefficient, which is the main dynamical attribute of interest here.

Reviewer #4 (Remarks to the Author):

The current study explores multi-proton dynamics to address the discrepancy between the experimentally observed rapid lateral proton mobility and its immobilization at an interfacial phosphate in MD simulations. Using Self-Consistent-Charge Density-Functional Tight-Binding with third-order energy correction (DFTB3), the authors simulate a membrane patch with one, two, or three protons. They systematically demonstrate how the trajectory of one proton significantly influences the others, effectively reconciling experimental and theoretical inconsistencies.

Comment 11: This is a compelling study, and I believe readers of Nature Communications will find it valuable. I have one minor comment: the pH conditions in the study seem quite acidic. How do these conditions compare to existing in vitro studies? Additionally, is it feasible to use bulk pH for comparison, considering that protons are released from the membrane and may not equilibrate with the bulk?

Response 11: The pH conditions were described in the 1st parag of the Methods/Classical MD Subsec. We now added that the pH is around 1, which incidentally is very similar to the pH in the injection pipette in Pohl's experiment. Our acidic pH conditions thus mimic experiment near the proton injection point.